# Gum Hydrocolloids Reinforced Silver Nanoparticle Sponge for Catalytic Degradation of Water Pollutants

**DOI:** 10.3390/polym14153120

**Published:** 2022-07-31

**Authors:** Rohith K. Ramakrishnan, Daniele Silvestri, Nechikkottil S. Sumitha, Nhung H. A. Nguyen, Karel Havlíček, Dariusz Łukowiec, Stanisław Wacławek, Miroslav Černík, Diwakar Tiwari, Vinod V. T. Padil, Rajender S. Varma

**Affiliations:** 1Institute for Nanomaterials, Advanced Technologies and Innovation (C × I), Technical University of Liberec (TUL), Studentská 1402/2, 461 17 Liberec, Czech Republic; rohith.kunjiparambil.ramakrishnan@tul.cz (R.K.R.); daniele.silvestri@tul.cz (D.S.); nhung.nguyen@tul.cz (N.H.A.N.); karel.havlicek@tul.cz (K.H.); stanislaw.waclawek@tul.cz (S.W.); miroslav.cernik@tul.cz (M.Č.); 2Department of Polymer Science and Rubber Technology, Cochin University of Science and Technology, Kochi 682 022, Kerala, India; sumithans93@gmail.com; 3Materials Research Laboratory, Faculty of Mechanical Engineering, Silesian University of Technology, Konarskiego 18 a St., 44-100 Gliwice, Poland; dariusz.lukowiec@polsl.pl; 4Department of Chemistry, Mizoram University (A Central University), Aizawal 796004, Mizoram, India; diwakarmzu@gmail.com; 5Regional Centre of Advanced Technologies and Materials, Czech Advanced Technology and Research Institute, Palacký University in Olomouc, Šlechtitelů 27, 783 71 Olomouc, Czech Republic

**Keywords:** biodegradation, biosponge, catalysis, green synthesis, silver nanoparticles, tree gum kondagogu

## Abstract

The accumulation of organic contaminants including dyes in aquatic systems is of significant environmental concern, necessitating the development of affordable and sustainable materials for the treatment/elimination of these hazardous pollutants. Here, a green synthesis strategy has been used to develop a self-assembled gum kondagogu-sodium alginate bioconjugate sponge adorned with silver nanoparticles, for the first time. The properties of the nanocomposite sponge were then analyzed using FTIR, TGA, SEM, and MicroCT. The ensued biobased sponge exhibited hierarchical microstructure, open cellular pores, good shape memory, and mechanical properties. It merges the attributes of an open cellular porous structure with metal nanoparticles and are envisaged to be deployed as a sustainable catalytic system for reducing contaminants in the aqueous environment. This nanocomposite sponge showed enhanced catalytic effectiveness (*k*_m_ values up to 37 min^−1^ g^−1^ and 44 min^−1^ g^−1^ for methylene blue and 4-nitrophenol, respectively), antibacterial properties, reusability, and biodegradability (65% biodegradation in 28 days).

## 1. Introduction

The release of a huge quantity of organic contaminants from the agricultural field, pharmaceutical, and chemical industries creates severe environmental pollution that directly causes a significant threat to the health of human beings and all other living organisms [1,2]. The treatment and removal of most pollutants in water have become incredibly challenging due to their water solubility and stability in the environment [3,4]. Although adsorption, biological treatment, solvent extraction, distillation, and chemical oxidation are some conventional methods for removing contaminants from water, catalytic hydrogenation is a highly efficient and convenient technique for degrading contaminants, producing fewer harmful effects [5]. For practical applications, the catalyst must meet the conditions of availability, low cost, high efficiency, reusability, and environmental friendliness [6].

Metal nanoparticles have garnered tremendous attention from researchers owing to their excellent selectivity and outstanding reactivity for catalytic destruction of contaminants [7]. Among them, silver nanoparticles (AgNPs) have been extensively applied as catalysts for reducing organic pollutants because of relatively abundant resources [8]. Capturing AgNPs on polymer templates minimizes the oxidation and aggregation of nanoparticles and simultaneously improves their catalytic activity [9]. The metal nanoparticles are well supported by an open-cellular sponge, which may be readily retrieved and reused after being utilized in catalytic reactions. Recently, NPs have been successfully decorated on chitosan, cellulose, and alginate-based materials for diverse applications [10,11]. However, these biopolymers require toxic chemicals, expensive equipment, and sophisticated technologies for extraction and purification [12]. In contrast, tree gums are non-toxic and sustainable alternatives which offer abundant functionalities, and advantages in terms of gel-forming capacity as well as degradability. Consequently, comprehensive research on these materials with promising features is necessary. An efficient and recyclable catalyst developed with a renewable material is environmentally and economically viable [10,11,12].

Gum kondagogu is a non-toxic polysaccharide extracted from the plant *Cochlospermum Gossypium* (Bixaceae family) [13]. It is a composition of arabinose, mannose, D-glucuronic acid, α-D galacturonic acid, β-D-galacturonic acid, α-D glucose, β-D-glucose, galactose, rhamnose, and fructose with an average molecular weight of 7.23 × 10^6^ g mol^−1^ [14]. It also contains proteins, tannin, and soluble fibers. Although template-assisted green synthesis may be conducted to generate nanoparticles making use of tree gums, their delicate nature makes three-dimensional fabrication and application problematic. On the other hand, gum functional groups can be coupled with other biopolymers to aid conjugate sponge formation with better characteristics and targeted application potential [15,16]. In this way, gum kondagogu combines with another polysaccharide, sodium alginate, derived from marine algae. Sodium alginate is a polysaccharide composed of two functional units β-D-mannuronic acid and α-L-guluronic acid joined by 1→4 bonds [17]. In addition, several free hydroxyl and carboxyl groups are distributed in the alginate structure, allowing them to form conjugate with other polymers. These conjugations convert the fragile and lower compressible tree gum sponge to a compressible composite sponge with structural integrity, hierarchical porosity, and low density [18]. Despite the benefits outlined above, only a few studies have been reported on gum sponges containing metal nanoparticles.

In the current work, AgNPs are green synthesized via an in-situ reduction process and uniformly decorated onto the gum kondagogu-sodium alginate conjugate sponge (Ag@KS) for the first time. The abundant functional groups in the gum and alginate facilitate conjugate formation and provide mechanical integrity. The remaining functionalities are further used for the green synthesis and stabilization of AgNPs to form a sustainable nanocomposite sponge. The reduction efficiency of the sponge for the nanoparticles’ synthesis has been assessed, and the optimum sponge-Ag precursor ratio has been identified for further examination. The physico-chemical and morphological characterization was then conducted to understand the properties of nanocomposite sponges. Both Gram-positive and Gram-negative bacteria were used to examine the antibacterial efficiency of the sponge.

Moreover, the post-degradation profile of the nanocomposite sponge shows its eco-friendly nature. The ligation of AgNPs onto the microporous sponge is anticipated to enhance the catalytic reduction of contaminants (4-Nitrophenol (4-NP) and Methyleneblue (MB) as examples) in an aqueous environment by allowing diffusion of polluted water to the interior of the porous network and at the same time preventing the AgNPs’ oxidation, aggregation, and leach out. The intriguing structural durability, good antibacterial and catalytic properties, and biodegradability would make it a rapid and sustainable solution to environmental pollution.

## 2. Materials and Methods

### 2.1. Materials

Gum kondagogu was supplied by Girijan cooperative society, Andhra pradesh, India. Sodium alginate, Silver nitrate (AgNO_3_, 99.8%), Calcium chloride (CaCl_2_, ≥93.0%), Sodium borohydride (NaBH_4_, ≥96.0%), Sodium hydroxide (NaOH, ≥98%), 4-Nitrophenol (4-NP, ≥99%), and Methylene blue (MB) were procured from Sigma-Aldrich, Darmstadt, Germany. The supernatant of the settled sludge from a municipal waste-water treatment facility (the Liberec WWTP, Czech R.) was used as an activated sludge for biodegradation experiments. All the chemicals purchased were of analytical grade and used as such without purification. In all the experiments, deionized water was used.

### 2.2. Preparation of Ag@KS Sponge

Gum kondagogu and sodium alginate were added in 1:1 ratio (*w*/*w*) to different concentrations of AgNO_3_ solution (4 mM, 6 mM, 8 mM, and 10 mM). The final polymer concentration in the resulting composite solution is maintained at 1.5 wt% [19]. The mixtures were then continuously stirred at 85 °C for 45 min to complete the reduction of nanoparticles under alkaline condition (pH = 10) using 0.1 N NaOH. The unreacted AgNO_3_ and excess NaOH after the reduction were removed through dialysis, the solutions were poured into the mould, and lyophilized at −52 °C for 24 h to obtain the Ag@KS sponges. Furthermore, post-synthesis crosslinking was performed using 5% CaCl_2_ solution, followed by washing with water to eliminate excess CaCl_2_, and lyophilized again. The pristine sample without AgNPs is referred to as KS sponge.

### 2.3. Characterization

UV–visible absorption spectra (UV-Vis) were recorded using UV Spectrophotometer (Hach DR 3900, Loveland, CO, USA) to monitor the reduction of Ag^+^ solution by the conjugate. The size and morphology of AgNPs were observed by transmission electron microscopy (TEM, Titan 80–300, FEI Company, Hillsboro, OR, USA). Silver concentration on sponge and leaching in water were evaluated using ICP-MS (Elan 6000, PerkinElmer, Akron, OH, USA). The X-ray diffraction (XRD) spectrum was recorded using a Rigaku MiniFlex, Austin, TX, USA, 600 X-ray powder diffractometer with a copper tube, Cu K_α_ (λ = 0.15406 nm), a tube voltage of 40 kV, and a current of 15 mA). Fourier transform infrared spectroscopy (FTIR) was performed on an infrared spectrophotometer (Nicolet iZ10, Thermo Scientific, Waltham, MA, USA), and thermal stability under a nitrogen atmosphere (at a flow rate of 50 mL min^−1^) was studied using Q500 Thermogravimetric Analyzer (TA instruments, New Castle, DE, USA) from 30 to 600 °C with a heating rate of 20 °C min^−1^. The morphology and porous architecture of sponges were observed by scanning electron microscope (SEM, Carl Zeiss, Ultra/Plus, Munich, Germany) equipped with energy dispersive X-ray spectroscopy (EDS). X-ray computed tomography (MicroCT, Skyscan 1272, Bruker, Billerica, MA, USA) was used to characterize the composite microstructure and porosity. A universal testing machine (Shimadzu Autograph AG-I series, Kyoto, Japan) was used to analyze the mechanical properties of the sponges in terms of compressive strength at a test speed of 5 mm min^−1^ for up to 80% compression. The sample dimensions were of 10 mm diameter and 20 mm length. The compressive strength was taken as the highest value of compressive stress, and the elastic moduli were calculated from the initial linear region (up to 1.5% strain) of the stress−strain curves. Bulk densities of the sponges were calculated from the weight (taken on an analytical balance SAB 124e, ADAM, Oxford, CT, USA) and volume of the sponges, where a digital calliper measured the dimensions of the sponges at five different positions.

### 2.4. Catalytic Degradation

Catalytic reduction of MB dye was performed with minor modifications of the previously reported procedure [20]. Briefly, 1.5, 1.9, and 3.1 mg of Ag@KS sponge were immersed separately in a mixture of NaBH_4_ (1 mL, 50 mM) and MB dye (4 mL, 30 μM) and stirred at room temperature for 60 min. The discoloration process was monitored using UV–Vis absorption spectroscopy at the MB maximum absorbance (λmax = 664 nm) at predefined time intervals of 0, 0.5, 1, 2, 3, 4, 5, 10, 15, 20, 30, 40, and 60 min.

Similarly, catalytic reduction of 4-NP was also accomplished as per the procedure reported earlier [21]. For this purpose, 1.0, 1.8, and 3.6 mg of Ag@KS sponge was immersed separately into an aqueous solution mixture of 4-NP (1 mM, 10 mL) and NaBH_4_ (0.3 M, 5 mL) at room temperature for 60 min. The reduction of 4-NP was monitored by UV–Vis spectroscopy at the 4-NP maximum absorbance (λ_max_ = 400 nm) at a predetermined time interval as that for MB.

The kinetics of the catalytic performance of the Ag@KS sponge were fitted to the pseudo-first order kinetic model. The kinetic rate constants were calculated using Equation (1):(1)$$ln\frac{A_{t}}{A_{0}}=-k_{\mathrm{app}}t$$
where *A*_0_ and *A_t_* are the measured absorbance at *t* = 0 and at time *t*, respectively, and *k*_app_ is the apparent rate constant of pseudo-first order kinetics.

For comparison of catalytic efficacy, the activity parameter (*k*_m_) was calculated according to Equation (2):(2)$$k_{m}=\frac{k_{\mathrm{app}}}{m}$$
where *m* is the total mass of the catalyst (g).

### 2.5. Reusability

The reusability of the Ag@KS Sponge for degradation of MB and 4-NP was investigated for five cycles. For MB, the first cycle was performed by immersing 1.0 mg of Ag@KS sponge in a mixture of NaBH_4_ (1 mL, 50 mM) and MB dye (4 mL, 30 μM), followed by stirring at room temperature for 60 min. From the second cycle onwards, the sponge was washed with water several times, freeze dried for the next cycle, and repeated the catalysis procedure for 60 min. A similar procedure was followed for 4-NP, where 1.0 mg of the Ag@KS sponge was immersed in a mixture of 4-NP (1 mM, 10 mL) and NaBH_4_ (0.3 M, 5 mL) at room temperature for 60 min.

The reusability of the sponge was calculated by means of UV-visible spectroscopy (664 nm for MB and 400 nm for 4-NP). The degradation percentage was calculated using Equation (3):(3)$$\mathrm{Degradation}\;\mathrm{of}\;\mathrm{pollutants}\;(\%)=\frac{C_{0}-C_{t}}{C_{0}}\times 100$$
where *C*_0_ is the initial concentration of pollutant, and *C_t_* is the pollutant concentration at time *t*.

### 2.6. Antibacterial Properties

The Gram-positive *Staphylococcus aureus* (CCM 3953) and Gram-negative *Escherichia coli* (CCM 3954) bacterial strains used in this analysis were procured from the Czech Collection of Microorganisms (Masaryk University, Brno, Czech Republic). Bacterial suspensions were freshly prepared in a nutrient broth (agar) by growing a single colony at 37 °C, overnight. The obtained bacterial culture was centrifuged into a pellet, which was further washed and resuspended in physiological solution (0.85% NaCl) and adjusted to an optical density of 0.1 at 600 nm. The bacteria were dispersed homogeneously on nutrient broth agar plates [23.5 g L^−1^, Plate Count Agar (M091), Himedia, India]. Sponges (KS and Ag@KS) were cut into a thickness of 3 mm and diameter of 5 mm, and placed onto the agar plates covered with bacteria. All the plates were inoculated for 24 h at 37 °C. The antibacterial activity was expressed in terms of zone of inhibition, which was determined as the total diameter (mm) of each sample plus the halo zone where bacterial growth was inhibited. The samples were tested in triplicates, and the average values are presented.

### 2.7. Biodegradation

The biodegradation of the KS and Ag@KS sponges in aqueous solutions under aerobic circumstances was evaluated using the technique described elsewhere [22]. The experiments were run in a Micro-Oxymax respirometer (Columbus Instruments International, Columbus, OH, USA). Approximately 2 g of the tested material were added to 95 mL of the biological medium (reference on ISO 14851:1999) and 5 mL of inoculum (activated sludge) into a 250 mL respiration cell and kept in a dark room at 20 ± 1 °C for 28 days. The oxygen consumption for an aerobic biodegradability was measured. The percentage of degradation was calculated based on the weight difference [23] using Equation (4):(4)$$\mathrm{Weight}\;\mathrm{loss}\;\left(\%\right)=\frac{W_{f}-W_{d}}{W_{d}}\times 100$$
where *W_f_* is the initial weight (g), while *W_d_* represents the residual weight of the sponge (g).

## 3. Results

### 3.1. Characterization of AgNPs

A green pathway for converting Ag^+^ ions to AgNPs and the fabrication of Ag@KS sponge is schematically represented in Figure 1. The negatively charged surface functional groups in the conjugate, particularly the carboxylic and hydroxyls groups, were exposed to the Ag^+^ precursor solution. These groups have high affinity and binding capability towards Ag^+^ ions through electrostatic interaction, resulting in Ag^+^ ions being reduced to AgNPs. The large number of active functional groups distributed along the backbone of the conjugate serve a dual role as a reducer and stabilizer for AgNPs.

A visible color change from off-white to yellowish-brown (Figure 2a,b) and UV-visible absorption spectra were used to monitor the conversion of Ag^+^ ions to AgNPs. The functional groups of the conjugate firmly bind and stabilize the nanoparticles, as observed from the highly colloidal and homogeneous solution after the formation of AgNPs. The color of the solution intensified with increasing AgNO_3_ concentration, indicating the generation of more AgNPs and their deposition on the surface of the conjugate at higher Ag precursor concentrations; similar findings have been reported previously [24,25]. The UV-visible spectra of AgNPs synthesized using varying AgNO_3_ concentrations are shown in Figure 2c, where increasing AgNO_3_ concentrations resulted in enhanced absorption peak strength at 420 nm. At 10 mM AgNO_3_ concentration, 90 ± 2% of Ag^+^ content in the precursor solution was retained in the Ag@KS sponge, as per the ICP-MS analysis (Appendix A). Similarly, 90% conversion of Ag^+^ ions to AgNPs has been reported in the green synthesis using plant leaf broth [26]. At concentrations higher than 10 mM, nanoparticle aggregation would compromise structural robustness and increase leaching. Therefore, 10 mM concentration was considered for further study as an optimal value for further experiments.

The predominantly spherical AgNPs in the size range of 1.5–6 nm are well disseminated in the conjugate solution, as shown by TEM imaging (Figure 2d). The particle size histogram of AgNPs, calculated using ImageJ software, shows the average diameter as 2.9 ± 0.5 nm (Figure 2e), which is lower than the AgNPs formed by porous lignocellulose hydrogel [27]. The lower size enhances the surface area and thereby catalytic performance. The functional groups on the conjugate serve as capping agents, preventing AgNPs from clumping together to ensure their uniform distribution on the sponge surface [28,29]. The SAED pattern (Figure 2f) implies a highly crystalline nature of nanoparticles, which was further confirmed by the XRD spectrum (Figure 2g). Four distinctive diffraction peaks were detected at 38.1°, 44.2°, 64.4°, and 77.2°, respectively, corresponding to (111), (200), (220), and (311) planes of face-centered cubic (fcc) crystal structure of metallic silver. The interplanar spacing (*d_h_ _k_ _l_*) values (2.352, 2.048 and 1.447 Å) determined from the XRD spectrum correlate well with the standard values [28,30]. The above results clearly show that the silver nanoparticles thus synthesized using the green method are nanocrystalline [31,32].

### 3.2. Ag@KS Sponge Characterization

The Ag@KS sponge developed by green synthesis has a low density (22 ± 0.5 mg/cm^3^) and 6.2 wt% silver content as AgNPs. The digital photographs of KS and Ag@KS sponges on a dandelion flower, exposing their low density, are represented in Appendix A.

Figure 3a shows the thermogravimetric analysis and the derivative curves of the KS and Ag@KS sponges. The initial (at temperature below 150 °C) weight loss of about 10% is associated with the desorption of physisorbed water on the samples. Both KS and Ag@KS sponges begin to disintegrate at around 200 °C, beyond which the thermograms take on different features. Because of the higher thermal stability conferred by strong interaction between AgNPs and the conjugate, the curve of Ag@KS is shifted towards the higher temperature. At 274 °C, the KS sponge degrades at a maximum rate, and the overall weight loss reached 38% of the initial weight. Ag@KS, on the other hand, exhibits a lower degradation rate and registers the highest degradation temperature (282 °C), and the corresponding weight loss is 31%. The thermogram of Ag@KS also presents a higher remnant weight of 40.6% at 600 °C compared to 35% for KS sample. This weight difference of 5.6% shows the silver content in the sponge. As in previously reported studies, Ag@KS sponges show better temperature stability than KS sponges due to the shielding effect induced by AgNPs [33].

FTIR spectra of KS and Ag@KS sponge are depicted in Figure 3b. Both sponges show peaks at 3280 cm^−1^, 1600 cm^−1^, and 1410 cm^−1^ corresponding to OH stretching, C=O stretching, and OH bending, respectively [34]. The intensity of these peaks in Ag@KS sponge declines due to the interactions between the conjugate and in-situ generated AgNPs [33]. The change of intensities confirms that the conjugate offers multiple sites to reduce Ag^+^ ions and simultaneously supports/stabilizes the ensued AgNPs.

The water-activated shape memory property of the tree gum conjugate sponge has not been well explored previously. The Ag@KS sponge exhibited excellent shape restoration while preserving its interconnected porous structure and physical integrity (Appendix A). It absorbs water and that could easily be squeezed out. The compact sponge takes up water again when immersed in it within a few seconds to return to its original shape and size. The complete shape recovery indicates that the solvent could rapidly and freely flow in and out of the material through the pores and open channels. Furthermore, the sponges from Ag@KS suspension could be customized to any shape and size based on the template container before freezing. The stability of the nanoparticles decorated on the Ag@KS sponge was also investigated using ICP-MS (Appendix A). The results demonstrated that, even after seven days of immersion in water, the loss of AgNPs is <0.2 wt% of the total silver content, which affirms the high capping capacity of the conjugate sponge surface.

The microporous structure of the KS sponge is obvious from SEM pictures (Figure 4a,b), which has not been altered by AgNPs (Figure 4c,d). The Ag@KS sponge has an open porosity of 92% (with an omissible closed porosity of 10^−4^%) with pore size ranging from 50–300 μm, as shown in the micro CT image and histogram (Figure 4e,f). During the swelling of the conjugate mixture, micro and nanosized cavities are formed that remain as such after nanoparticle reduction. During the time of freeze-drying, water molecules were sublimed, leaving a porous spongy network with a mean pore size of 170 μm (Figure 4f). A prominent peak of silver is observed from the EDX spectrum of Ag@KS, and its homogeneous distribution is evident from the elemental mapping (Appendix A). It allows the entering pollutant to be highly exposed to the AgNPs on the sponge surface, thus resulting in enhanced catalytic efficiency.

Figure 5 depicts the compressive stress–strain curves of the KS and Ag@KS sponges. The stress–strain curves of these porous structures show three stages: elastic region, plateau region, and densification. The initial deformation step occurs when cell walls are elastically compressed until they cease to collapse. This linear region (up to 1.5% strain) determines the elastic modulus, 10 kPa and 45 kPa for the KS sponge and Ag@KS sponge, respectively. The presence of AgNPs enhances the stiffness of the cell walls of the Ag@KS sponge and hence increases the elastic modulus. Cells are collapsing under strain near the beginning of the plateau zone. In the Ag@KS sponge, the slope towards the end of the plateau region has risen slightly compared to the KS sponge. The completion of cell collapse and the contacting of opposite cell walls result in sponge densification, the final deformation step [35]. The AgNPs distributed along the sponge surface improved the resistance to cell wall breaking, creating stronger cells. In addition, crack propagation and wall straining would be delayed, resulting in increased failure strength.

The homogeneous tree–gum conjugate formation and subsequent reduction and stabilization of AgNPs lead to a highly regular porous nanocomposite sponge that is more favorable than the irregular topology of many of the earlier reported sponges, for example, AgNPs embedded chitosan-alginate sponge [36]. According to Tang et al., oriented open cells enhance the mechanical robustness of the pore structure [37]. Altogether, AgNPs on the conjugate sponge increase the compressive strength to a greater extent.

### 3.3. Catalytic Reduction of MB and 4-NP

The model contaminants, 4-NP and MB dye, were used to evaluate the catalytic potential of the Ag@KS sponge. These pollutants have a strong absorption peak in visible light, making them easier to detect by UV-visible spectroscopy. Hence, they are commonly deployed for evaluating the catalytic potential of different materials [38,39].

The linear dependency of the logarithm of absorbance on time confirmed the catalytic degradation of both toxic organic dyes following the pseudo-first-order kinetics (Equation (1)). As shown in Figure 6a, MB’s apparent rate constants (*k*_app_) are 0.041, 0.058, and 0.116 min^−1^ for 1.5, 1.9, and 3.1 mg of sponge, respectively. An increasing tendency could be observed in the *k*_app_ values as the catalyst amount increased (Figure 6c). For comparing catalytic performance, the activity parameter of *k*_m_, which equals 37 min^−1^ g^−1^, was used, which is the ratio of rate constant to the weight of catalyst (3.1 mg) [40,41]. The *k*_m_ parameter was compared with other catalysts reported in the literature (Appendix A), which revealed that the Ag@KS sponge has more excellent MB conversion rates than most catalysts used.

When KS sponge was utilised, no catalytic reduction of MB to leucomethylene blue (LMB) was detected (data not shown), indicating the potential of AgNPs as a catalyst. The reusability of catalyst is of important concern in the environmental and economic perspectives and, therefore, the reusability of Ag@KS sponge for MB degradation was investigated. The results showed retention of catalytic activity over five cycles, though a slight reduction (78% degradation from 98%) was observed (Figure 6e). As previously reported by Kamal et al., the decrease in pollutant degradation during repeated cycles could be attributed to the catalyst poisoning by S atoms [42].

Figure 6b shows the catalytic reduction of 4-NP, which follows pseudo-first-order kinetics with apparent rate constants of 0.044, 0.074, and 0.105 min^−1^ for 1.0, 1.8, and 3.6 mg of sponge, respectively. Here, the *k*_app_ values continued to rise as the catalyst amount increased (Figure 6d), and the *k*_m_ values of 44 min^−1^ g^−1^ are calculated from 1.0 mg weight of catalyst. In this case too, a comparison of the activity parameter (*k*_m_) with that of other documented sponges in the literature (Appendix A) indicates that Ag@KS performed better for catalytic degradation of 4-NP to 4-Aminophenol (4-AP) amongst many reported examples. The reusability of Ag@KS sponge for 4-NP catalytic reduction was also determined (Figure 6f), wherein, after five cycles, 75% of pollutants are degraded, demonstrating steady catalytic performance and remarkable durability of the Ag@KS sponge, endowed by strong capping capacity of the functional groups of the conjugate.

### 3.4. Catalysis Mechanism of Ag@KS Sponge

Figure 7 depicts the proposed pathway for contaminants removal using the Ag@KS sponge. The sponge has many micropores and open channels, while AgNPs are immobilized on the sponge surface walls. The presence of AgNPs induces BH_4_^−^ (of NaBH_4_) oxidation with a release of electrons, resulting in a negatively charged layer on the sponge’s surface [41]. When the contaminant molecules (MB or 4-NP) contact the charged sponge surface, AgNPs facilitate the electron relay from BH_4_^−^ to MB/4-NP, leading to the disruption of pollutant’s chromophore structure. The 4-NP is reduced to 4-Aminophenol and MB to leucomethylene blue.

### 3.5. Antibacterial Properties

The antibacterial activity of KS and Ag@KS sponges were evaluated against *Escherichia coli* and *Staphylococcus aureus* by measuring the zone of inhibition of bacterial growth around discs placed on agar medium (Figure 8). Figure 8a,c show no inhibition zone around the KS sponge in both bacterial media. In contrast, the Ag@KS sponge showed a prominent zone of inhibition of diameter 11.2 mm ± 0.2 mm and 11.0 mm ± 0.2 mm, respectively, for *Escherichia coli* and *Staphylococcus aureus*, indicating excellent antibacterial activity (Figure 8b,d). AgNPs with a size of less than 10 nm have the ability to penetrate bacterial cell walls and subsequently change the structure and cause cell damage [43]. Overall, our findings indicate that AgNPs decorated on this sustainable sponge have a distinct inhibitory effect on the microbial growth of *Staphylococcus aureus* (Gram-positive) and *Escherichia coli* (Gram-negative), thus widening their applications in the environmental field [44].

### 3.6. Biodegradation of the Sponge

The biodegradability of the sponges was tested under aerobic conditions with an inoculum taken from WWTP. Figure 9 shows the biochemical oxygen demand (BOD) for KS and Ag@KS sponges. Aerobic microbial species disintegrate the sponges into smaller molecules during the degradation process through metabolic or enzymatic activities. In the case of the KS sponge, after the initial phase without BOD, the degradation profile goes linearly upwards. For Ag@KS, the reluctant phase is significantly longer (to around 150 h), and the slope of the degradation phase is also smaller. Hence, the KS degradation rate is higher than that of Ag@KS. After 28 days (672 h), the final BOD value for the KS sponge is 1566 ± 16 mg O_2_ L^−1^, while that for the Ag@KS sponge is 535 ± 35 mg O_2_ L^−1^. By comparing the weight loss, 90% of KS and 65% of Ag@KS were degraded in 28 days. The rapid deterioration of the KS sponge implies that the microorganisms present in the sludge could act on it faster, but the presence of AgNPs on the Ag@KS sponge slows down the activity of microorganisms in the material and thereby reduces the rate of degradation.

The bio-based nanocomposite sponges have gained popularity as a greener choice for reducing waste generation and lowering production costs. The results presented here suggest that the green synthesized sponges can be biologically degraded by the microbes, thus minimizing the secondary contamination of the environment.

## 4. Discussion

In this study, we have developed a conjugate of gum kondagogu and sodium alginate that overwhelms the components individually in terms of hydrophilicity and mechanical strength. In addition to being an affordable and renewable biopolymer source, tree gum conjugate simultaneously aids the generation and stabilization of AgNPs with greater than 90% retention of NPs. The AgNPs immobilized conjugate sponge with low density and high porosity are then crosslinked with Ca^2+^ ions. The structural stability, surface morphology, and compressive strength can be tailored both by the decoration of AgNPs on the sponge and ionic crosslinking. As the NPs are evenly distributed along the surface of the Ag@KS sponge, the oxidation, aggregation, and leach out of AgNPs are effectively minimized, leading to enhanced efficiency towards the catalytic reduction of MB and 4-NP from aqueous media. When compared to previously published catalyst systems, the nanocomposite sponge used in this study demonstrated impressive catalytic activity. The shape recovery property of the sponges in water is of much significance since it enables the repeated usage of the material without compromising structural integrity. The reusability study of the sponge showed retention of catalytic activity of 78% and 75% for MB and 4-NP, respectively, even after five cycles. Additionally, it effectively reacted with both Gram-positive and Gram-negative bacteria during the antibacterial study. Finally, the biodegradability of the sponge (65% within 28 days) affirms its eco-friendliness and at the same time offers numerous opportunities for environmental remediation applications.

## 5. Conclusions and Future Perspectives

In conclusion, AgNPs immobilized on tree gum based sustainable sponges represent a new class of catalyst carrier systems. The preparation of Ag@KS sponges is facile with the deployment of renewable and earth-abundant materials and can therefore be converted to many other green synthesized nanoparticle-based catalyst systems with excellent catalytic efficiency. The combination of high macroporosity (92%) and low density (22 ± 0.5 mg/cm^3^) allows the effective mass transfer of the contaminants to the catalytic sites (AgNPs). The observed negligible leaching of Ag nanoparticles (<0.2 wt %) from Ag@KS sponge supports its use for long-term water remediation operations. The novelty of this model sponge catalyst is the exploitation of greener synthesis and biodegradability aspects in the design of newer sustainable materials. Furthermore, they display excellent antibacterial properties against *Escherichia coli* and *Staphylococcus aureus.* The catalysis was achieved with an activity parameter of 37 min^−1^ g^−1^ and 44 min^−1^ g^−1^ for MB and 4-NP, respectively. The prepared Ag@KS sponge shows excellent reusability even after five cycles of usage. These findings will pave the way to exploit the enormous potential of tree gum-based sponge systems in a vast variety of applications in environmental, industrial, and medicinal fields.

## Figures and Tables

**Figure 1 polymers-14-03120-f001:**
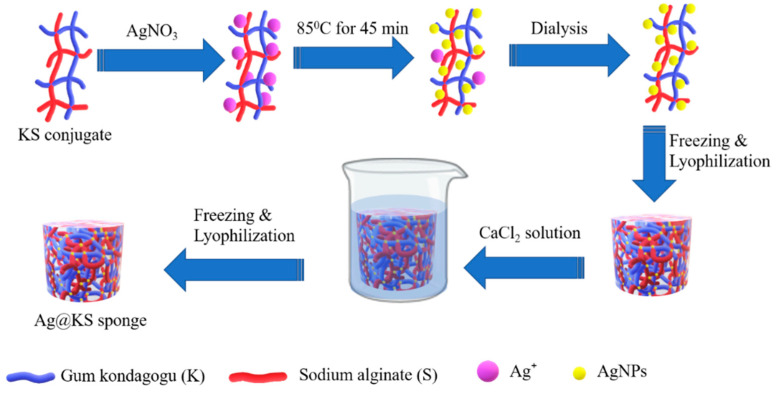
Schematic illustration of the Ag@KS sponge fabrication.

**Figure 2 polymers-14-03120-f002:**
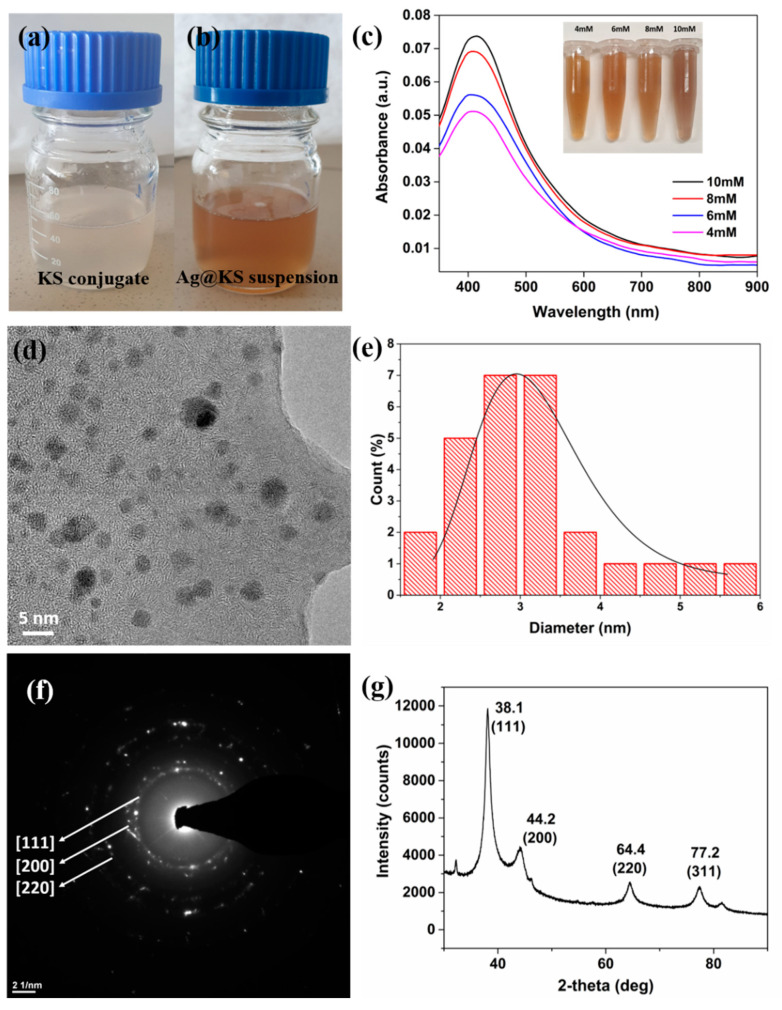
(**a**) Digital photograph of KS conjugate solution and (**b**) Ag@KS suspension, (**c**) UV–visible spectra of the AgNPs synthesized from various concentrations of AgNO_3_, the insert is a photograph of Ag@KS suspension at various AgNO_3_ concentrations, (**d**) TEM image of Ag@KS suspension at 10 mM AgNO_3_, (**e**) histogram representing the particle size distribution for 10 mM AgNO_3_ precursor, (**f**) SAED pattern, and (**g**) XRD spectrum of Ag@KS sponge.

**Figure 3 polymers-14-03120-f003:**
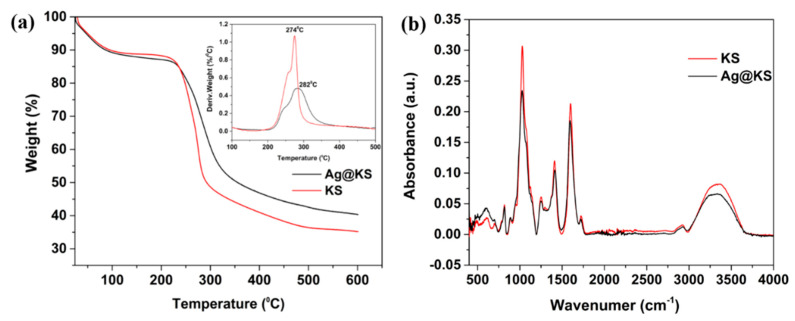
(**a**) Thermogravimetric profile of KS and Ag@KS sponge (derivative curve is inserted), (**b**) FTIR spectra of KS and Ag@KS sponge.

**Figure 4 polymers-14-03120-f004:**
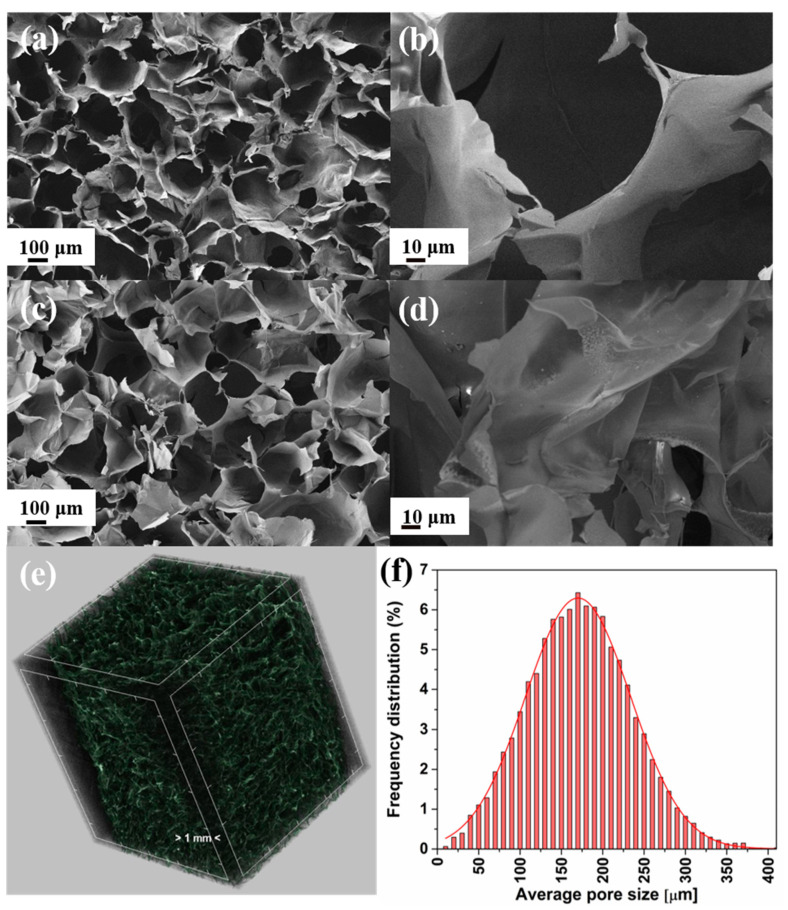
(**a**,**b**) SEM micrographs of KS sponge and (**c**,**d**) Ag@KS sponge at different magnifications, (**e**) 3D micro-CT image of Ag@KS sponge, (**f**) the pore size distribution of the Ag@KS sponge.

**Figure 5 polymers-14-03120-f005:**
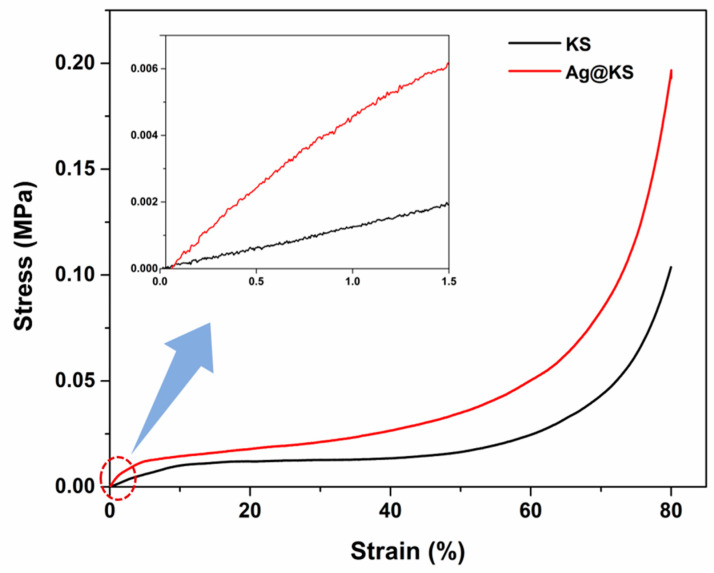
The compressive stress–strain curves of the KS and Ag@KS sponges (the initial linear region is inserted).

**Figure 6 polymers-14-03120-f006:**
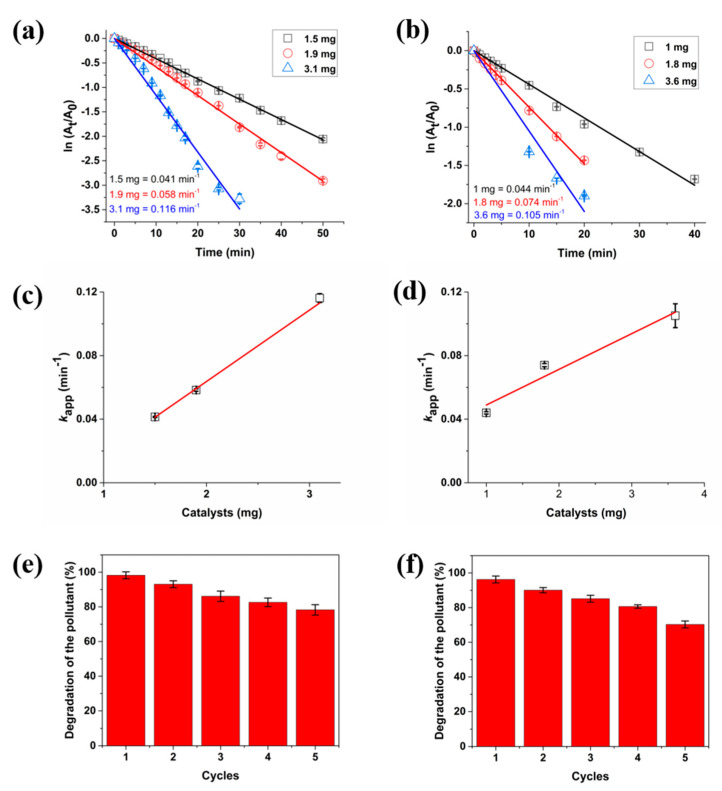
The pseudo-first-order kinetic curves of the catalytic reduction of (**a**) MB and (**b**) 4-NP, *k*_app_ values of (**c**) MB and (**d**) 4-NP under different Ag@KS concentrations; and degradation percentage of (**e**) MB and (**f**) 4-NP over five reduction cycles.

**Figure 7 polymers-14-03120-f007:**
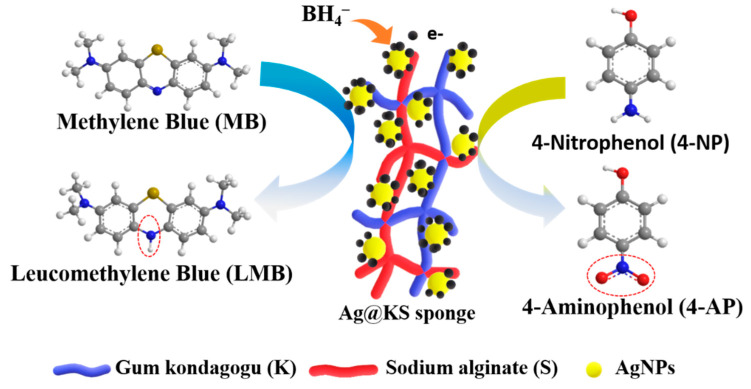
Proposed catalytic reduction mechanism of MB and 4-NP in the Ag@KS sponge network, using NaBH_4_.

**Figure 8 polymers-14-03120-f008:**
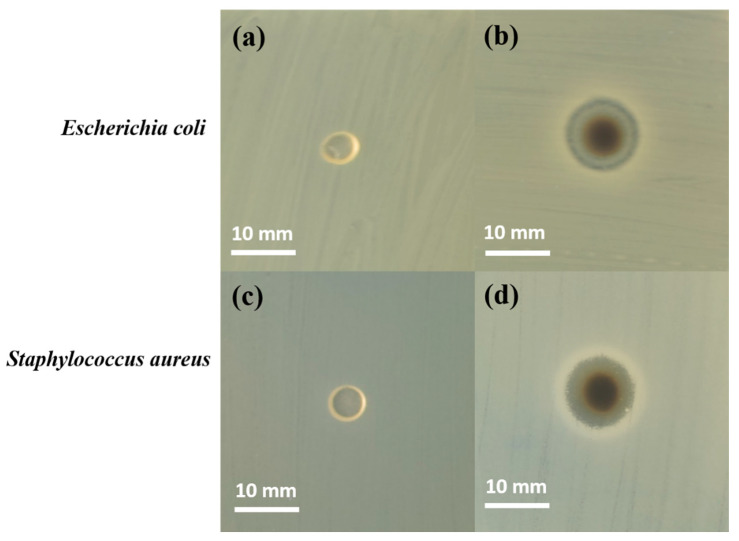
Formation of inhibition zone around KS (**a**,**c**) and Ag@KS (**b**,**d**) sponge discs against *E. coli* and *S. aureus* after 24 h incubation.

**Figure 9 polymers-14-03120-f009:**
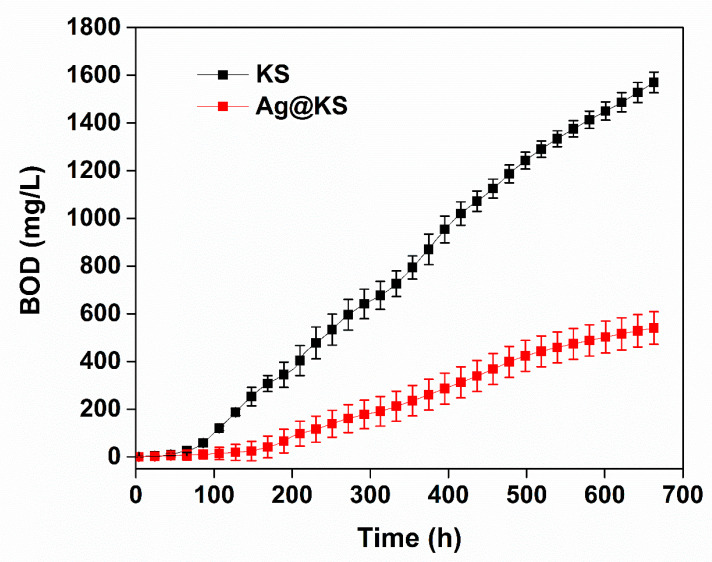
Biochemical oxygen demand of KS and Ag@KS sponges for a period of 28 days of enzymatic degradation.

## Data Availability

Not applicable.

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
