# Peer review of "Gum Hydrocolloids Reinforced Silver Nanoparticle Sponge for Catalytic Degradation of Water Pollutants"

_polymers, 2022, doi:10.3390/polym14153120_

Round 1

Reviewer 1 Report

This manuscript reports a novel green malachite gum kondagogu-sodium alginate bioconjugate sponge synthesis strategy decorated with silver nanoparticles. This nanocomposite sponge shows enhanced catalytic effect. Surprisingly, it also has biodegradability to avoid secondary environmental pollution. The results and topics are interesting. However, some points of the manuscript should be improved. Specific comments are given below.

1.    The preparation of a self-assembled peacock glue-sodium alginate bioconjugate sponge, how it self-assembles, please explain in detail.

2.    , Silver nanoparticles were prepared using a mixture of malachite gum in preparation of Ag@KS sponge, sodium alginate, and AgNO3 solution at 85°C. Why choose 85°C? Will the reaction temperature be too high to destroy the hydroxyl and carboxyl groups of polysaccharides?

3.    In preparation of Ag@KS sponge, after the Ag@KS sponge was lyophilized in a mold and then using 5% CaCl2 solution for post-synthesis crosslinking, what role does Ca2+ play here?

4.    How about the stability of AgNPs?

5.    In 3.1, 90±2% of the Ag+ content in the precursor solution is retained in the Ag@KS sponge at the concentration of 10 mM AgNO3. When the concentration is less than 10 mM, will Ag+ be 100% reduced to AgNPs? AgNO3 exceeds 10 mM concentration, will aggregation of AgNPs form. The concentration and results of AgNPs aggregation should be supplemented in the article.

6.    What is the mechanism of the Ag@KS sponge catalyzing MB and 4-NP? Please elaborate.

Reviewer 2 Report

I reviewed the article with the title ``Gum hydrocolloids reinforced silver nanoparticle sponge for catalytic degradation of water pollutants``.  The article topic is intriguing and promising in the area. Overall, the article structure and content are suitable for the Polymers journal. I am pleased to send you major-level comments, there are some serious flaws that need to be corrected before publication. Please consider these suggestions as listed below.

·         The title seems good, but the abstract seems to be wired. Please add one more introductory line of your objective at beginning of the abstract. Highlight the core finding.

·         Keywords are ok

·         Research gap should be delivered in a clearer way with the directed necessity for future research work.

·         Introduction section must be written in a more quality way, i.e., more up-to-date references addressed. Please target the specific gap such as 2015-2021 etc.

·         The novelty of the work must be clearly addressed and discussed, compare previous research with existing research findings, and highlight novelty.

·         Several lumpy references are there, please take a strong look in revision.

·         Please remove the reference 1-4 and simply cite this one article- Role of nanomaterials in the treatment of wastewater: a review.

·         Please remove the reference 12,13 and simply cite this one article- Recent advances in metal decorated nanomaterials and their various biological applications: a review.

·         What is the main challenge? Why author choose this material? Please highlight this in the introduction part.

·         Please remove the reference 14,15 and simply cite this one article- Silver nanoparticles: various methods of synthesis, size affecting factors and their potential applications–a review.

·         Please cite this single article and remove references 17-22. Umar, K.; Yaqoob, A.A.; Ibrahim, M.; Parveen, T.; Safian, M. Environmental applications of smart polymer composites. Smart Polym. Nanocompos. Biomed. Environ. Appl. 2020, 15, 295–320.

·         The main objective of the work must be written in the clearer and more concise way at the end of the introduction section.

·         Please provide space between numbers and units. Please revise your paper accordingly since some issue occurs in several spots in the paper. 

·         Please check the abbreviations of words throughout the article. All should be consistent.

·         Please include all chemical/instrumentation brand names and other important specifications.

·         Please add chemical reagents brand specifications.

·         Regarding the replications, the authors confirmed that replications of the experiment were carried out. However, these results are not shown in the manuscript, how many replicates were carried out by experiment? Results seem to be related to a unique experiment. Please, clarify whether the results of this document are from a single experiment or from an average resulting from replications. If replicated were carried out, the use of average data is required as well as the standard deviation in the results and figures shown throughout the manuscript. In the case of showing only one replicate explain why only one is shown and include the standard deviations.

·         Please remove the dot (full stop) at the end of the word Figure. It’s wrongly formatted. Please check.

·         Discussion on part is not sufficient, please take a sincere round. Please add a comparative profile as well.

·          Please add a comparative profile section to compare your results and prove how it is better than previous ones.

·         Section 5 should be renamed by Conclusion and Future perspectives. The conclusion section is missing some perspective related to the future research work, quantifying the main research findings, and highlighting the relevance of the work with respect to the field aspect.

·         To avoid grammar and linguistic mistakes, Major level English language should be thoroughly checked. Please revise your paper accordingly since several language issues occur in several spots in the paper.

·         Reference formatting needs careful revision. All must be consistent in one formate. Please follow the journal guidelines.

Reviewer 3 Report

In my opinion the paper with title: Gum hydrocolloids reinforced silver nanoparticle sponge for 2 catalytic degradation of water pollutants could be published in the journal Polymers if the authors will make the following corrections and specifications suggested:

The article needs major revision.

At Chapter 2.2. Preparation of Ag@KS sponge, please, explain clear the method of preparation Ag@KS. Isn't clear when is it maintained the concentration of final polymer concentration at 1.5 wt%. What means the polymers in this content?

It is not clear how the experiments were conducted.

At Chapter 2.4. Catalytic degradation, please present more clearly how the experiments were conducted for MB and 4-NP. How the sample was taken, what volume of sample was taken, how long did you take the samples, and which is the frequency of sampling.

At Chapter 2.5. Reusability, please explain in the text what is the duration of operation of the catalyst into a cycle. Please enter in the paper the mathematical relationship used to calculate the efficiency and reduction efficiency of  the catalyst after 1-5 cycles of use. It is not clear how the experiments were conducted.

At Chapter 2.6. Antibacterial properties please introduce in the paper the name of the nutrient used for microorganism growing and specify how the antibacterial action of the catalyst has been demonstrated according to the results showed in Chapter Result, 3.5. Antibacterial properties. It is not clear how the experiments were conducted.

At Chapter 2.7. Biodegradation, please explain clear haw the percentage of biodegradability was calculated. It is not clear how certain parameters were calculated

In the Figure 5 was showed the variation of Stress (MPa) with Strain, %, please explain what represent the Strain, and the relationship using to calculated it. It is not clear how certain parameters were calculated

At chapter 3.3. Catalytic reduction of MB and 4-NP in Figure 6 on Oy axis are showed the variation of ln At/A0, please explain the signification of this terms in experiments performed.

Round 2

Reviewer 1 Report

The authors have addressed the problem very well, and the manuscript can be accepted in the present form.

Reviewer 2 Report

I re-review the revised version. It is ready for publication in its present form. 

Reviewer 3 Report

In my opinion the paper with title: “Gum hydrocolloids reinforced silver nanoparticle sponge for 2 catalytic degradation of water pollutants” can be published in the journal Polymers. The authors made the corrections and specifications suggested.